# Disposition of Cefquinome in Turkeys (*Meleagris gallopavo*) Following Intravenous and Intramuscular Administration

**DOI:** 10.3390/pharmaceutics13111804

**Published:** 2021-10-28

**Authors:** Mohamed Elbadawy, Ahmed Soliman, Amira Abugomaa, Adel Alkhedaide, Mohamed Mohamed Soliman, Mohamed Aboubakr

**Affiliations:** 1Department of Pharmacology, Faculty of Veterinary Medicine, Benha University, Moshtohor, Toukh 13736, Qalioubiya, Egypt; mohamed.aboubakr@fvtm.bu.edu.eg; 2Pharmacology Department, Faculty of Veterinary Medicine, Cairo University, Giza 12211, Egypt; galalpharma@cu.edu.eg; 3Faculty of Veterinary Medicine, Mansoura University, Mansoura 35516, Dakahliya, Egypt; 4Clinical Laboratory Sciences Department, Turabah University College, Taif University, Taif 21995, Saudi Arabia; a.khedaide@tu.edu.sa (A.A.); mmsoliman@tu.edu.sa (M.M.S.)

**Keywords:** cefquinome, bioavailability, HPLC, pharmacokinetics, poultry, turkeys

## Abstract

The bioavailability and pharmacokinetics in turkeys of cefquinome (CFQ), a broad-spectrum 4th-generation cephalosporin antibiotic, were explored after a single injection of 2 mg/kg body weight by intravenous (IV) and intramuscular (IM) routes. In a crossover design and 3-weeks washout interval, seven turkeys were assigned for this objective. Blood samples were collected prior to and at various time intervals following each administration. The concentration of CFQ in plasma was measured using HPLC with a UV detector set at 266 nm. For pharmacokinetic analysis, non-compartmental methods have been applied. Following IV administration, the elimination half-life (t_1/2ʎz_), distribution volume at steady state (Vd_ss_), and total body clearance (Cl_tot_) of CFQ were 1.55 h, 0.54 L/kg, and 0.32 L/h/kg, respectively. Following the IM administration, CFQ was speedily absorbed with an absorption half-life (t_1/2ab_) of 0.25 h, a maximum plasma concentration (C_max_) of 2.71 μg/mL, attained (T_max_) at 0.56 h. The bioavailability (F) and in vitro plasma protein binding of CFQ were 95.56% and 11.5%, respectively. Results indicated that CFQ was speedily absorbed with a considerable bioavailability after IM administration. In conclusion, CFQ has a favorable disposition in turkeys that can guide to estimate optimum dosage regimes and eventually lead to its usage to eradicate turkey’s susceptible bacterial infections.

## 1. Introduction

Pathogenic bacteria constitute a menace to the health of humans and animals and lead to a significant economic burden; thus, an antibacterial intervention is a vital consideration [1]. However, the emergence of bacterial resistance to antibacterials is a constant medical issue due to the regular and hazardous use of classical antibiotics. This problem can partly be solved by the usage of appropriate and effective antibacterial drugs. 

In veterinary practice, cephalosporins are very often prescribed for the remediation of bacterial infections [2,3]. Their mode of action is attributed to the disruption of the peptidoglycan layer which is important for the structural integrity of the bacterial cell wall resulting in lysis and death of the bacterial cell [2]. Cefquinome (CFQ) is a broad-spectrum parenteral bactericidal cephalosporin (4th-generation) antibiotic. It was developed specifically for veterinary usage [4]. The unique structural design of CFQ (Figure 1) was obtained by setting a methoxyimino-aminothiazolyl moiety into cephalosporin’s acyl side chain and a quaternary quinoline group at position 3 of the cephem ring resulting in the zwitterionic structure of CFQ. This unique design notably boosts the antibacterial efficacy, spectrum, and stability of CFQ against β-lactamases-producing and methicillin-resistant bacteria as *Enterococci* and *Staphylococci* [5,6,7]. Further, the zwitterionic structure facilitates its rapid permeation beyond biological membranes and through the porins of bacterial cell walls, ensuring its speedy effect following injection [8]. These special structure grants preferable pharmacokinetic properties to CFQ such as speedy absorption, high bioavailability, primary elimination in an unchanged form via the kidneys, and low protein binding [9]. The spectrum of CFQ includes different bacterial strains such as *Streptococci*, *Staphylococci*, *Enterobacteriaceae family* (*E. coli*, *Salmonella*, *Klebsiella*, *Citrobacter* species), and *Pseudomonas aeruginosa* [10,11,12]. Therefore, it is prescribed for the remediation of bacterial diseases of the respiratory system in equines and poultry, as well as acute mastitis, mastitis-metritis-agalactia syndrome, and foot rot disease in bovines [2,10,13,14]. 

Extensive pharmacokinetic studies have been performed for CFQ in numerous mammals under normal and different physiological experimental conditions such as foals [14], ponies [15], horses [16], pigs [17], piglets [18,19,20], dogs [21,22], calves, [23,24], cattle [25,26], sheep [27,28], goats [29,30], rabbits [31,32] and mice [33,34]. The CFQ pharmacokinetics have been also determined in fishes including tilapia [35], carp [36], and salmon [37]. These investigations established a conceptual framework for the reasonable and clinical usage of CFQ in these species. A limited amount of research has been conducted to determine the pharmacokinetics of CFQ in avian species such as ducks [9], chickens [38], ducklings and goslings [39], black swans (*Cygnus atratus*) [11], and laying hens [40]. The data of these studies revealed differences among these avian species in CFQ disposition, entailing precise assessment before its clinical usage in new avian species. Usage of allometric scaling for dose extrapolation among avian species could lead to inaccuracies, especially when drugs are prone to metabolic alterations [41,42].

The kinetics of CFQ has not been studied in turkeys yet. Therefore, the current study aimed to determine the CFQ’s disposition profile in turkeys following a single intravenous (IV), and intramuscular (IM) administration to get information for the future establishment of optimal dosage regimens.

## 2. Materials and Methods

### 2.1. Reagents and Chemicals

Cefquinome sulfate powder of 84.1% purity was purchased from Hebei Yuanzheng Pharmaceutical Co., Ltd. (Hebei, China). The injectable solutions of CFQ were prepared by dissolving CFQ powder in sterilized distilled water. An HPLC grade CFQ standard (95% purity) was purchased from Fujifilm Wako Pure Chemical Co. (Osaka, Japan), and used for validation of the calibration method. The purity percentage of CFQ sulfate powder (84.1%) and CFQ standard (95%) was considered when preparing the administered doses and the solutions of CFQ standard. Other reagents and chemicals used in the present study were of analytical reagent grade. 

### 2.2. Experimental Birds

Seven male turkeys, 15 weeks of age, 6–7 kg body weight (BW), were purchased from a local commercial turkey farm. The turkeys were maintained at 20 °C, 65% relative humidity, and a 12 h/day light cycle. There were two weeks of acclimatization before experimentation to ensure that their bodies are free from any residual drugs such as antibacterials and anticoccidials. Standard commercial pelleted feed, free from antibiotics and coccidiostats, and water were supplied *ad libitum*. The health status of turkeys was daily assessed, and no signs of disease were seen. The experimental protocol was developed following the directions of the Declaration of Helsinki and accepted by the Ethical Committee of the Faculty of Veterinary Medicine, Benha University, Egypt (Number: BUFVTM 02-07-21). All efforts were exerted to maintain comfort and minimize pain to turkeys.

### 2.3. Experimental Design

The doses were calculated precisely based on the purity percentage of CFQ sulfate powder (84.1%) and the individual BW of each turkey prior to CFQ injection. A crossover design [43] was used where the washout interval was three weeks (Figure 2). The turkeys were given a single dose of 2 mg/kg BW of CFQ either IV into the left brachial vein or IM into the thigh muscles. The selected dose in the current study was based on the doses that were mostly used in avian species, chicken, duck, and geese [9,38,39,40]. Blood samples (1.5 mL) from each turkey were gathered from the right brachial vein into tubes containing heparin just before medication (time = 0) and then at 0.08, 0.16, 0.25, 0.5, 1, 2, 4, 6, 8, and 12 h after CFQ injection. After centrifugation of these blood samples for 10 min at 1600× *g*, plasma samples were separated into 1.5 mL tubes, labeled appropriately, and stored at −30 °C until it was assayed by HPLC system.

### 2.4. Analytical Method 

The concentration of CFQ in turkey’s plasma was analyzed by HPLC as determined previously [9]. Briefly, 300 µL each plasma sample was deproteinized using 600 µL of methanol, vortexed for 10 s and centrifuged at 20,000× *g* for 15 min in a cooling centrifuge at 4 °C. Thereafter, the supernatant of each sample was gathered, dried with nitrogen gas, reconstituted in 300 µL of mobile phase, and filtered through a 0.45-μm HPLC filter (Chromatodisc^®^, 4P, Kurabo Biomedical Industries, Ltd., Osaka, Japan). The HPLC system (HP-1100 HPLC system, Agilent Technologies, Palo Alto, CA, USA) used in the present study was equipped with a degasser, a quaternary pump, an autosampler, a UV detector, and a column heater. The mobile phase was composed of sodium perchlorate (85 mM (9.16 gm/L))-triethylamine (4.6 mL, to dissolve sodium perchlorate)-phosphate (to set pH at 3.6)-buffer and acetonitrile (88:12, *v*/*v*). The flow rate was put at one mL/min. The analytical separation of CFQ was done by a reverse phase C18 Hypersil-BDS column (250 × 4.6 mm, 5 µm, Agilent Technologies) with the temperature maintained at 30 °C, a UV wavelength set at 266 nm, and the injection volume was 20 µL.

### 2.5. Method Validation

The stock solution of CFQ (equivalent to 1 mg/mL) was prepared by dissolving 10.53 mg of CFQ standard (95%) in 10 mL of the mobile phase. The working solutions of the CFQ standard (0.01, 0.05, 0.1, 1, 2.5, and 5 µg/mL) by diluting the CFQ stock solution with mobile phase. Calibration was performed by spiking 60 µL of each standard working solution to 240 µL of blank turkey’s plasma and assayed as described above. The means of five values were applied to plot standard curves (peak area vs. CFQ concentration). The recovery rate from plasma, inter-day and intra-day coefficients of variability were determined via repetitive analysis of plasma samples. The average plasma recovery rate of CFQ was high (94%). The values of intra- and inter-day coefficients of variability were ranged from 4.07~4.63% and 4.91~5.35%, respectively (*n* = 5, three times, three days). The lower limits of detection (LOD) and quantification (LOQ) were determined using the signal-to-noise method and were 0.010 and 0.035 µg/mL, respectively. The peak of CFQ in plasma has appeared on a chromatogram at 12 min (Appendix A). 

### 2.6. Plasma Protein Binding Extent of CFQ

The extent of CFQ binding to plasma protein was assessed in vitro using the ultrafiltration method [44]. The various CFQ standard concentrations (equivalent to 0.2, 2, 20 and 100 µg/mL were prepared in deionized water by diluting the stock solution of CFQ (equivalent to 1 mg/mL deionized water). Thereafter, 50 µL of each standard working solution was mixed in triplicate with 950 µL of fresh blank turkeys’ plasma to yield plasma samples spiked with known CFQ concentrations of 0.01, 0.1, 1, and 5 µg/mL. After vortexing for 20 s, the spiked samples were maintained for 30 min at 37 °C to permit binding of CFQ with plasma proteins of plasma samples. Subsequently, one mL of each spiked sample was dropped into the reservoir of Ultrafree^®^ centrifugal filter tube (pore diameter of 0.45 μm, Millipore Corporation, Tokyo, Japan). The ultrafiltration was conducted by centrifugation of the Ultrafree^®^ centrifugal filter tubes at 2500× *g*, at 37 °C for up to 30 min until the required amount of the ultrafiltrate was achieved. In the obtained ultrafiltrate, CFQ concentrations were assayed as mentioned above. The percentage of CFQ plasma protein binding was assessed regarding the initial sample concentration using the following Equation (1):(1)$$\mathrm{Plasma}\;\mathrm{protein}\;\mathrm{binding}\;\left(\%\right)=100-\left[\left\{\frac{\mathrm{concentration}/\mathrm{mL}\;\mathrm{ultrafiltrate}\;}{\;\mathrm{initial}\;\mathrm{concentration}/\mathrm{mL}\;\mathrm{plasma}\;}\right\}\times 100\right]$$

### 2.7. Pharmacokinetic Analysis

After IV and IM administrations and HPLC analysis of plasma samples, the calculated plasma concentrations of CFQ for each turkey at each sampling time were presented as mean ± standard deviation (SD) values (Appendix A). The non-compartmental analysis method [45] was applied to calculate several pharmacokinetic parameters using the concentration vs. time data of CFQ in each turkey following the IV and IM administrations and the WinNonlin program (version 6.1, Pharsight, Mountain View, CA, USA). The calculated parameters include total body clearance (Cl_tot_), distribution volume at steady state (Vd_ss_), and mean residence time (MRT). The trapezoidal method was applied to calculate the area under the concentration-time curve (AUC), the area under the first moment curve (AUMC). Mean residence time (MRT) was calculated as MRT = AUMC/AUC and the total body clearance (Cl_tot_) as Cl_tot_ = Dose/AUC. The elimination rate constant (ʎ_z_) was calculated by linear regression of several points (4–6) on the terminal phase of logarithmic plasma concentrations vs. time curve. The terminal half-lives (t½_ʎz_) were calculated using where t_½ʎz_ = 0.693/ʎ_z_. The peak plasma level (C_max_) and time to C_max_ (T_max_) after IM administration of CFQ were determined using the curve of plasma concentration vs. time for each turkey. The absolute bioavailability (F) of CFQ after IM was calculated as F = AUC_IM_/AUC_IV_ × 100. The half-life of absorption of CFQ after IM administration was determined using the equation: t_1/2ka_ = 0.693 × mean absorption time (MAT). All obtained values are expressed as Mean ± SD.

## 3. Results

No abnormalities such as irritation, pain signs, or lameness were noticed in turkeys following the administrations of CFQ. The semi-logarithmic plasma concentration vs. time curves of CFQ after the single IV and IM administration of 2 mg/kg BW are illustrated in Figure 3 and the mean ± SD of pharmacokinetic parameter values are shown in Table 1. 

The individual pharmacokinetic parameters are presented in Appendix A. Following IV administration, CFQ concentrations declined from 5.25 ± 0.0613 µg/mL at 0.083 h to 0.012 ± 0.001 µg/mL at 12 h post-administration (Table 1). The half-life interval of elimination (t_1/2ʎz_), Vd_ss_, and Cl_tot_ of CFQ were 1.56 ± 0.0631 h, 0.547 ± 0.0133 L/kg, and 0.323 ± 0.0255 L/h/kg, respectively. 

After IM administration, CFQ was speedily absorbed and detectable in plasma at 5 min of administration with a half-life of absorption (t_1/2ab_) of 0.253 ± 0.022 h, a peak plasma level (C_max_) of 2.71 ± 0.061 μg/mL and attained (T_max_) at 0.558 ± 0.007 h. The bioavailability (F) of CFQ was 95.6 ± 0.676%. No CFQ was detected in any turkey’s plasma 24 h after IV or IM administration. The in vitro protein binding extent of CFQ spiked at concentrations of 0.01 to 5 µg/mL ranged from 10% to 13%. The mean protein binding percentage at different drug concentrations was 11.6% (Table 2). 

## 4. Discussion

Analyzing the absorption, distribution, metabolism, and excretion (ADME) profile of drugs is the chief goal of pharmacokinetic studies. A drug’s pharmacokinetic profile following a single injection may be well represented by C_max_, T_max_, t_1/2ʎz,_ and AUC evaluation [46]. Differences among avian species in the disposition of CFQ were shown in laying hens [40], healthy ducks [9], ducklings and gosling [39], healthy chickens [38], and black swans [11], necessitating thorough pre-clinical evaluation before administration to new species as turkeys. 

As far as we know, the present research is the first to check the pharmacokinetic profile of CFQ in turkeys. We checked the disposition profile of CFQ in healthy turkeys after administration of 2 mg/kg BW via the IV and IM routes to obtain data for the future establishment of appropriate dosage regimens. The obtained data revealed that CFQ has rapid and almost complete absorption and satisfactory elimination rate that allows reasonable dosing intervals in turkey. Further, there were no noticed tissue irritation, lameness, or pain symptoms. Similarly, there were no adverse events following the administration of CFQ to several avian species such as ducklings and goslings [39], black swans (*Cygnus atratus*) [11], laying hens [40], healthy chickens [38], and mammals such as goats [47], buffalo calves [23] and horses [16]. These data from different species elucidate the safety of CFQ.

In the current study, we compared the obtained CFQ pharmacokinetic data in turkeys with those of CFQ in other avian species and mammals (Table 3). Our data revealed that after IV administration, CFQ was eliminated speedily from the turkey’s plasma with a t_1/2ʎz_ of 1.56 h. These data is quite identical to that reported for CFQ in ducks (1.57 h [9]) and black swans (*Cygnus atratus*, 1.69 h [11]), longer than that of CFQ in chickens (1.29 h [38]), ducklings (0.97 h [39]), and shorter than that of CFQ in gosling (1.73 h [39]). In mammals, CFQ showed also speedy elimination after IV administration. The t_1/2ʎz_ of CFQ were 2.32 h in horses [16], 2.1 h in cattle [26], 1.85 h in premature calves [24], 0.72 and 0.93 h in rabbits [31] and [32], 0.98 h beagle dogs [22] and 0.78 h in sheep [27]. In foals and mares, the t_1/2ʎz_ of CFQ after IV administration of 1 mg/kg was 1.82 and 2.33 h, respectively [15]. These data indicate the rapid elimination of CFQ after IV administration in turkeys, other avian species, and most mammals. However, in fish species, the t_1/2ʎz_ of CFQ was substantially longer after intraperitoneal administration of 10 mg/kg BW. The t_1/2ʎz_ of CFQ was 6.88 h in crucian carp (*Carassius auratus gibelio*) at 25 °C [36], 6.05 h in tilapia (*Oreochromis niloticus*) at 30 °C [35], and 20.6 h in coho salmon (*Oncorhynchus kisutch*) at 10 °C [37]. This prolonged elimination of CFQ in fish species compared to other species has been demonstrated also for other antibacterials in fish. These data indicate there are variations in the elimination rate of CEQ among various species. 

The Vd_ss_ value is used to indicate the amount of drug in the body under equilibrium conditions. It is the proportionality constant between plasma concentrations of drugs and their total amount in the body [48]. In the present study, the obtained Vd_ss_ for CFQ in turkeys was 0.547 L/kg. this value is close to that recorded for CFQ in ducks (0.41 L/kg [9]), chickens (0.49 L/kg [38]), laying hens (0.871 L/kg [40]), black swans (0.32 L/kg [11]), and gosling (0.43 L/kg [39]). These findings demonstrated that the tissue distribution of CFQ in turkeys and other avian species is quite limited after IV administration. In mammals, the Vd_ss_ rate of CFQ was also limited after IV administration. The Vd_ss_ of CFQ were 0.36 L/kg in horses [16], 0.28 L/kg h in cattle [26], 0.37 L/kg in premature calves [24], 0.21 and 0.26 L/kg in rabbits [31,32], 0.30 L/kg beagle dogs [22], 0.28 L/kg [28] and 0.36 L/kg [27] in sheep. In foals and mares, the Vd_ss_ of CFQ after IV administration of 1 mg/kg was 0.09 L/kg [14], and 0.22 L/kg [15], respectively. In fish species, the Vd/F (volume of distribution corrected for bioavailability) of CFQ was also limited after intraperitoneal administration of 10 mg/kg BW. The values of 0.2 and 0.33 L/kg were recorded in crucian carp [36] and tilapia [35], respectively. These data of limited distribution indicate the limited penetration of CFQ to the intracellular compartment after IV administration in turkeys, and most other avian, mammals, and fish species. This could be referred to that CFQ is an organic acid with a hydrophilic nature (low lipophilicity), and a small pKa value (2.51–2.91) [49,50]. In the present study, the in vitro plasma protein binding of CFQ was limited (11.5%) as has been shown in previous studies (ranged from 5–15% in most species [40]). The protein binding assay used in the present study did not account for the non-specific drug binding to the centrifugal device which may be a limitation in the present study. 

The obtained value of total clearance (CL_tot_) of CFQ from turkey bodies in the current study was 0.323 L/h/kg. Similar values were reported for CFQ in chickens (0.35 L/h/kg; [38]), duckling (0.32 L/h/kg; [39]). However, this value was shorter than that of CFQ in gosling (0.45 L/h/kg; [39]), laying hens (0.62 L/h/kg [40]), and longer than CFQ in ducks (0.22 L/h/kg; [11]) and black swans (0.13 L/h/kg [11]). In mammals, the systemic clearance of CFQ was also small after IV administration of 2–4 mg/kg BW. The CL_tot_ of CFQ were 0.158 L/h/kg in horses [16], 0.12 L/h/kg h in cattle [26], 0.13 L/h/kg in premature calves [24], 0.18 and 0.25 L/h/kg in rabbits [31,32], 0.24 L/h/kg beagle dogs [22], 0.16 L/h/kg in sheep [27]. In foals and mares, the CL_tot_ of CFQ after IV administration of 1 mg/kg was 0.18, and 0.13 L/h/kg, respectively [15]. All these findings indicate the rapid elimination of CFQ after IV administration in turkeys and most other avian species and mammals. In fish species, the rate of systemic clearance of CFQ was reported after intraperitoneal administration of 10 mg/kg BW. The values of 0.020 and 0.037 L/h/kg were reported in crucian carp [36] and tilapia [35], respectively, indicating a slower clearance of CFQ in fishes than in avian species and mammals. These data indicate there are variations in the clearance of CEQ among various species. 

Following IM administration to turkeys, CFQ was speedily absorbed as the measured absorption half-life t_1/2ab_ was short (0.25 h). The absorption seems slower in turkeys compared with ducks (t_1/2ab_: 0.12 h, [9]), black swans (t_1/2ab_: 0.12 h, [11]) and chickens (t_1/2ab_: 0.07 h [38]). In mammals, the t_1/2ab_ values of CFQ were also short after IM injection.

**Table 3 pharmaceutics-13-01804-t003:** Pharmacokinetic parameters of cefquinome (CFQ) in different species after intravenous (IV) and intramuscular (IM) administration of 2 mg/kg BW except for duck (5 mg/kg BW).

Species	Chickens		Ducks		Black Swans		Ducklings		Gosling	
Dose (mg/kg)	2		5		2		2		2	
Route	IV	IM	IV	IM	IV	IM	IV	IM	IV	IM
β (1/h)	0.54 ± 0.04	0.53 ± 0.08	0.44 ± 0.02	0.39 ± 0.03	42.09 ± 0.09	0.43 ± 0.03	-	-	-	-
t_1/2α_ (h)	0.43 ± 0.19	0.58 ± 0.27	0.19 ± 0.05	0.46 ± 0.30	0.31 ± 0.03	-	0.019	0.343	0.446	0.483
t_1/2β_ (h)	1.29 ± 0.10	1.35 ± 0.20	1.57 ± 0.06	1.79 ± 0.11	1.69 ± 0.85	1.62 ± 0.11	0.972	1.717	1.737	1.403
AUC_0–∞_ (μg·h/mL)	5.33 ± 0.55	5.13 ± 1.06	25.12 ± 2.31	23.78 ± 3.87	16.5 ± 4.92	12.17 ± 4.32	6.248	4.220	4.396	5.008
Vd_ss_ (L/kg)	0.49 ± 0.05	-	0.41 ± 0.0	-	0.32 ± 0.17	-	0.042	-	0.432	-
Cl_tot_ (L/kg/h)	0.35 ± 0.04	-	0.22 ± 0.02		0.13 ± 0.04		0.320	-	0.455	-
t_1/2ab_ (h)	-	0.07 ± 0.02	-	0.12 ± 0.02	-	0.12 ± 0.04	-	-	-	
C_max_ (µg/mL)	-	3.04 ± 0.71	-	9.38 ± 1.61	-	5.71 ± 1.43	-	4.010	-	3.400
T_max_ (h)	-	0.25 ± 0.06	-	0.38 ± 0.06	-	0.39 ± 0.19	-	0.163	-	0.203
F (%)	-	95.81 ± 5.81	-	93.28 ± 13.89	-	74.2 ± 26.3	-	67.5	-	113.9
Reference	[38]		[9]		[11]		[39]		[39]	
**Species**	**Pigs**		**Rabbits**		**Sheep**		**Dogs**		**Calves**	
**Dose (mg/kg)**	**2**		**2**		**2**		**2**		**2**	
**Route**	**IV**	**IM**	**IV**	**IM**	**IV**	**IM**	**IV**	**IM**	**IV**	**IM**
β (1/h)	-	-	0.76 ± 0.11	0.69 ± 0.13	-	-	0.72 ± 0.11	0.84 ± 0.13	0.40 ± 0.11	0.15 ± 0.02
t_1/2α_ (h)	0.30 ± 0.08	1.33 ± 0.42	-	-	0.06 ± 0.04	0.31 ± 0.05	0.12 ± 0.05	-	-	-
t_1/2β_ (h)	2.32 ± 0.47	4.92 ± 1.14	0.93 ± 0.14	1.04 ± 0.22	0.78 ± 0.19	1.88 ± 0.40	0.98 ± 0.14	0.85 ± 0.15	1.85 ± 0.44	4.47 ± 0.69
AUC_0–∞_ (μg·h/mL)	18.35 ± 5.32	17.22 ± 4.11	11.08 ± 4.06	10.40 ± 1.23	5.83 ± 0.45	5.19 ± 0.25	8.51 ± 1.27	8.24 ± 0.80	15.74 ± 3.57	22.75 ± 6.18
Vd_ss_ (l/kg)	-	-	0.21 ± 0.03	-	0.36 ± 0.06	-	0.30 ± 0.03	-	0.37 ± 0.10	-
Cl_tot_ (L/kg/h)	0.12 ± 0.03	-	0.18 ± 0.05	-	0.34 ± 0.03	-	0.24 ± 0.03	-	0.13 ± 0.03	-
t_1/2ab_ (h)	-	0.24 ± 0.05	-	-	-	0.31 ± 0.05	-	0.14 ± 0.05	-	-
C_max_ (µg/mL)	-	3.36 ± 0.54	-	8.87 ± 2.07	-	2.60 ± 0.14	-	4.83 ± 0.79	-	4.56 ± 0.75
T_max_ (h)	-	0.83 ± 0.28	-	0.25 ± 0.12	-	0.50 ± 0.00	-	0.43 ± 0.11	-	1.00 ± 0.00
F (%)	-	85.13 ± 9.93	-	95.23 ± 9.84	-	89.31 ± 6.06	-	97.8 ± 9.40	-	141.22
Reference	[51]		[31]		[27]		[22]		[28]	

β: elimination rate constant, t_1/2α_; distribution half-life after IV injection, t_1/2β_; elimination half-life after IV injection, AUC_0-∞_; area under plasma concentration-time curve from zero time to infinity, Vd_ss_; volume of distribution at steady-state, Cl_tot_; total body clearance, t_1/2ab_: absorption half-life after IM administration, C_max_; maximum plasma concentration, T_max_; time to peak plasma concentration, F; absolute bioavailability.

The values were 0.28 h in pigs [52], 0.31 h in sheep [27], 0.14 h in beagle dogs [22], 0.16 h in buffalo calves [53], and 0.45 h in foals [14]. In fish species, the values were 0.04 h crucian carp [36] and 0.028 h in tilapia [35]. These data demonstrated that it takes little time for CFQ to enter the systemic circulation and establish an efficient plasma level in turkeys, other avian species, mammals, and fish species. The calculated t_1/2ʎz_ of CFQ in turkeys after IM administration was 1.71 h, longer than that of IV administration (1.56 h). The longer t_1/2ʎz_ after extravascular administration than after IV administration may result from precipitation of the drug at the injection site or flip-flop phenomenon, in which the absorption rate of a drug is slower than its rate of elimination [31,54,55]. In the present case, the MAT value is expected to be longer than the MRT value after IV administration (MRT_IV_) as demonstrated before in rabbits [31]. However, the shorter MAT value (0.374 h) than MRT_IV_ (1.7 h) in the present study does not support the flip-flop phenomenon. After IM injection, some drugs precipitate in increasing amounts at the injection site to provide increasing values of the t_1/2ʎz_ [54,56]. Therefore, the precipitation of CFQ at the IM injection site might be the reason for its longer t_1/2ʎz_ compared with the t_1/2ʎz_ after IV administration. The calculated t_1/2ʎz_ of CFQ in turkeys after IM administration is remarkably identical to the value reported for CFQ in ducks (1.79 h [9]), duckling (1.71 h [39]), black swans (1.62 h [11]), and longer than CFQ in chickens (1.35 h [38]) and gosling (1.40 h [39]). However, it is shorter than that obtained for CFQ in laying hens (2.22 h) after IM administration at a dose of 5 mg/kg BW [40]. In mammals, the t_1/2ʎz_ values of CFQ after IM injection were variable as 4.44 h in pigs [52], 4.85 h in goats [47], 0.85 h in beagle dogs [22], 4.74 h in premature calves [24], 3.73 h in buffalo calves [53], 0.45 h in foals [14], 1.04 [31] and 0.72 h [32] in rabbits. In fish species, the values were longer as 7.39 h in crucian carp [36] and 5.81 h in tilapia [35]. 

The C_max_ of CFQ in turkeys was 2.71 ± 0.161 µg/mL achieved at (T_max_) 0.560 ± 0.0181 h. The obtained C_max_ and T_max_ value of CFQ (2 mg/kg) in turkeys was lower than those of CFQ administered at the same dose in duckling (4.01 µg/mL achieved at 0.163 h [39]), goslings (3.40 µg/mL achieved at 0.203 h [39]), and chickens (3.04 µg/mL achieved at 0.25 h [38]). In mammals, different values for C_max_ and T_max_ of CFQ after IM administration of 2 mg/kg were also demonstrated. The values were 6.43 ± 0.637 µg/mL achieved at 0.78 ± 0.771 h in pigs [52], 2.37 ± 0.13 µg/mL achieved at 1 h in goats [47], 4.83 ± 0.79 µg/mL achieved at 0.43 ± 0.11 h in beagle dogs [22], 4.56 ± 0.75 µg/mL achieved at 1 h in premature calves [24], 8.78 ± 2.07 µg/mL achieved at 0.25 ± 0.12 h in rabbits [31], and 6.93 ± 1.72 µg/mL achieved at 0.33 ± 0.12 h [32] in rabbits. 

The calculated systemic bioavailability of CFQ after the IM administration to turkeys was 95.6%, which is almost identical to that reported for CFQ at a dose of 2 mg/kg BW in chickens (95.8% [38]) and ducks at a dose of 5 mg/kg BW (93.3% [9]), higher than for CFQ at a dose of 2 mg/kg BW in duckling (67.5% [39]), laying hens at a dose of 2 mg/kg BW (66.8% [40]), and black swans at a dose of 2 mg/kg BW (74.2% [11]). However, it is lower than for CFQ at a dose of 2 mg/kg BW in goslings (113.9% [39]). In mammals, the bioavailability of CFQ after IM administration was also high as it were 97.8% in beagle dogs [22], 141.2% in premature calves [24], 86.3% in buffalo calves [53], 95.2% in rabbits [31]. After subcutaneous administration in sheep the value was also high (123.5% [28]). These data show that CFQ is rapidly and nearly completely absorbed following IM administration to turkeys, other avian species, and mammals. This might be most likely owed to its zwitterionic nature, which allows CFQ to permeate easily into the biological membranes [8]. High bioavailability of CFQ after IM administration into thigh muscles may suggest that the renal portal system and tubular excretion play an insignificant role in the potential first-pass effect often associated with the administration to the caudofemoral portion of the body in birds and reptiles [57]. However, the lack of comparative data on CFQ administration into another part of the body (e.g., breast muscle) prevents any firm conclusions. This may be considered another limitation of the study.

These differences in the disposition profile of CFQ among species are common and attributed to several factors such as inter-species variation in the extent of metabolism, differences in the assay methods used, given dose, blood sampling times, age of animals, and/or the health status [58].

The antibacterial activity of CFQ is considered to be time-dependent [4,26,59]. This means that the antibacterial efficacy of CFQ is proportionally related to the time that free CFQ concentration in plasma surpasses the MIC (minimum inhibitory concentration) for certain pathogens during the inter-dosing interval (%T > MIC) [60,61,62]. The field usage of CFQ in turkeys has not been established yet due to the paucity of pharmacokinetic studies and the MICs data of CFQ for turkey’s pathogenic bacterial strains. However, MIC values of ≤ 0.1 µg/mL typically reported for pathogens isolated from different species suggest the high clinical efficacy of this antibiotic [63,64]. In literature, data revealed that CFQ is effective against numerous bacteria isolated from other poultry species that can induce health problems in turkeys. For example, *E. coli* O78 and *Salmonella* strain C79-13 from chickens were very susceptible to low serum concentrations of CFQ with a MIC of 0.063 and 0.25 µg/mL, respectively [65]. Also, *Pasturella multocida* and *Ornithobacterium rhinotracheale* isolated from avian species like turkeys, ducks, geese, chicken, and pheasants were very sensitive to CFQ [66,67]. Further, *Riemerella anatipestifer* isolated strains from ducks and geese were susceptible to CFQ with MIC_50_ of 0.031 µg/mL and MIC_90_ levels of 0.5 µg/mL [68]. Additionally, in black swans, the MIC_50_ and MIC_90_ values for CFQ against 49 *E. coli* strains were 0.063 and 0.5 µg/mL, respectively [11]. In the current study, the time that plasma level of CFQ maintained above 0.1 µg/mL after the IM administration of 2 mg/kg BW was 8 h. In general, to attain an adequate therapeutic efficiency of numerous cephalosporins, the time in which free drug concentration in plasma surpasses the MIC should be higher than 40% of the inter-dosing interval [48,69,70,71,72]. Based on these data and the findings of the present study, a twice-daily dosage of 2 mg/kg BW of CFQ given intramuscularly to turkeys at the age of 17–20 weeks and 7–8.5 kg of BW would be efficient against several susceptible bacterial strains. However, some recent reports on antimicrobial pharmacokinetics in turkeys revealed that elimination processes differ depending on the age of the birds. For example, Poźniak et al. found that the rapid growth in turkeys significantly affected amoxicillin pharmacokinetics wherein younger turkeys (2 kg), the t_1/2ʎz_ was approximately two-fold shorter (0.67 h) than at 12 kg (1.28 h) [73]. Also, Świtała et al. reported in turkeys that, between the 5th and 15th week of age, CL_tot_ of metronidazole declined from 3.6 to 1.2 mL/min/kg causing a twofold rise in the MRT and t_1/2ʎz_ [74]. These differences are probably due to the changes that occurred in heart rate, cardiac output, enzymatic functions, and or alteration in clearing organ perfusion [74]. Therefore, the lack of the perspective of age-dependent change in cefquinome pharmacokinetics is a limitation of the present study.

## 5. Conclusions

After IM administration of CFQ, there were no local reactions and adverse effects. Further, CFQ revealed rapid absorption and high bioavailability. Concentrations exceeding MIC values for most of the poultry pathogens indicate that the repeated (twice-daily) IM administration of CFQ at 2 mg/kg BW might be highly efficacious against susceptible bacterial pathogens in turkeys. However, additional studies should be carried out to set a multiple dosage regimen, assess the clinical efficacy of CFQ and its residues in the edible tissues of this species.

## Figures and Tables

**Figure 1 pharmaceutics-13-01804-f001:**
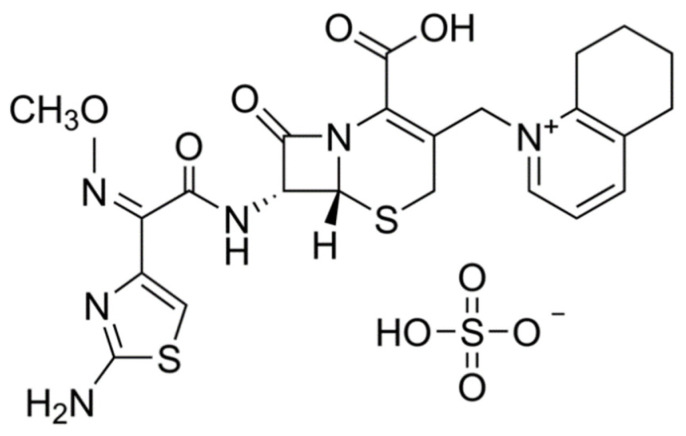
Chemical structure of cefquinome.

**Figure 2 pharmaceutics-13-01804-f002:**
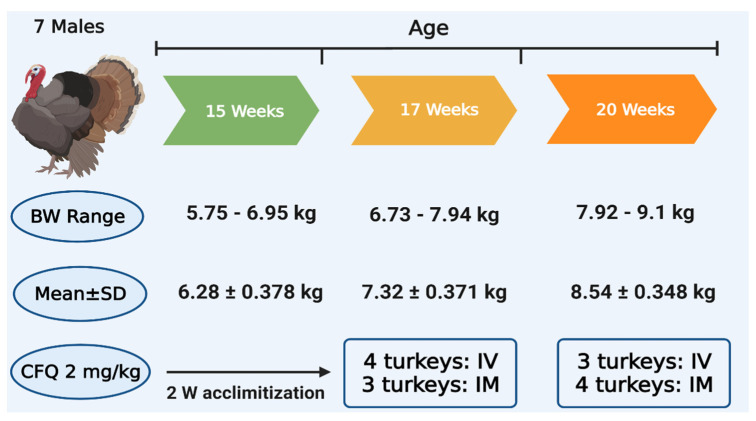
Body weights and experimental design for injection of CFQ to turkeys.

**Figure 3 pharmaceutics-13-01804-f003:**
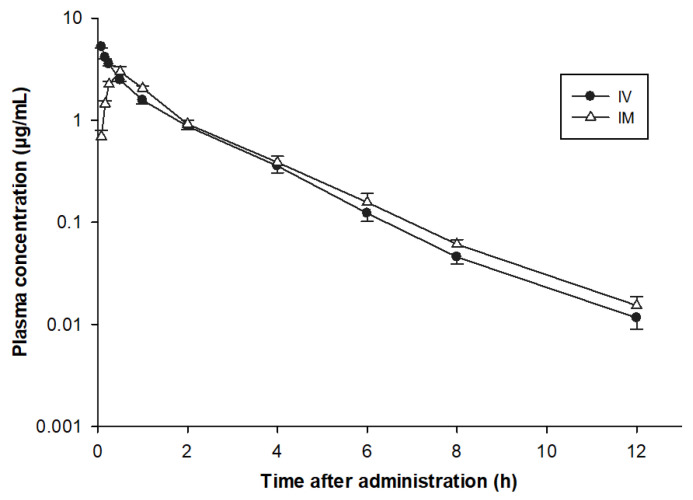
Semi-logarithmic graph depicting the time-concentration of cefquinome in plasma of turkeys after a single intravenous (IV) (●) and intramuscular (IM) (△) administration of 2 mg/kg BW (*n* = 7).

**Table 1 pharmaceutics-13-01804-t001:** Plasma pharmacokinetic parameters of cefquinome in turkeys following intravenous (IV) and intramuscular (IM) administration of 2 mg/kg BW (*n* = 7). All values are expressed as Mean ± SD.

Parameters	Unit	IV	IM
C^0^	µg/mL	6.63 ± 0.302	—
t_1/2ab_	h	—	0.253 ± 0.0580
t_1/2ʎz_	h	1.56 ± 0.0631	1.71 ± 0.076
AUC_0–__∞_	μg·h/mL	6.22 ± 0.428	5.94 ± 0.443
AUMC_0–__∞_	μg·h/mL	10.6 ± 1.23	12.3 ± 1.52
MRT	h	1.70 ± 0.082	2.07 ± 0.117
MAT	h	—	0.374 ± 0.062
Vd_ss_	L/kg	0.547 ± 0.0133	—
Cl_tot_	L/kg/h	0.323 ± 0.0255	—
C_max_	µg/mL	—	2.71 ± 0.161
T_max_	h	—	0.558 ± 0.018
F	%	—	95.6 ± 1.78

C^0^; concentration at zero time (immediately after single IV injection), t_1/2ab_; absorption half-life after IM administration, t_1/2ʎz_; terminal elimination half-life, AUC_0–∞_; area under plasma concentration-time curve from zero time to infinity, AUMC_0–∞_; area under moment curve from zero time to infinity, MRT; mean residence time, MAT; mean absorption time, Vd_ss_; volume of distribution at steady-state, Cl_tot_; total body clearance. C_max_; maximum plasma concentration, T_max_; time to peak plasma concentration, F; absolute bioavailability.

**Table 2 pharmaceutics-13-01804-t002:** In vitro plasma protein binding percentage of cefquinome in turkey plasma (*n* = 3).

Fortified CFQ Concentrations (µg/mL) in Blank Plasma	Protein Binding %	SD
0.01	13.01	1.78
0.1	11.9	0.853
1	11.3	1.48
5	10.1	1.12
Mean	11.5	1.31

## Data Availability

All relevant data are included in the manuscript.

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
