# Peer review of "Disposition of Cefquinome in Turkeys (Meleagris gallopavo) Following Intravenous and Intramuscular Administration"

_pharmaceutics, 2021, doi:10.3390/pharmaceutics13111804_

Round 1

Reviewer 1 Report

This manuscript aims at describing the pharmacokinetics of cefquinome, a cephalosporin antibiotic, in turkeys. The topic is interesting and relevant to veterinary science and the manuscript is properly structured and comprehensible. However, it has several shortcomings that require attention before publication may be recommended by this Reviewer. Below is the list of general and more specific issues that need to be addressed by the Authors:

  1. In the Introduction the Authors claim that: “This problem [antimicrobial resistance] can partly be solved by continual development and evaluation of new and effective anti-bacterial drugs.” This Reviewer is not feeling comfortable with this statement as it seems to suggest that the primary solution to the development and spread of antimicrobial resistance is simply to use different and new drugs. We know it is not true as the introduction of new antimicrobial groups to mass treatment of animals (e.g. fluoroquinolones) has been associated with the rise in resistance that may be dangerous not only to animals but also to humans. However, cefquinome has been developed solely for veterinary purposes and this may be a better argument in this reasoning. I would like to ask the Authors to rephrase this statement to reflect the need for the prudent use of antimicrobials.
  2. The Authors ignore the recent papers on antimicrobial pharmacokinetics in turkeys that revealed that elimination processes will be different depending on the age of the birds. This is a substantial lack as the guidelines formulated at the end of the discussion may only be valid if cefquinome is administered to turkeys at the age of 15 weeks and 6-7 kg of body weight. Based on studies on another beta-lactam antibiotic (amoxicillin) we may expect that in younger turkeys (e.g. 5 weeks, 2 kg) the elimination half-life will be approx. two-fold shorter and the Authors’ recommendations would lead to suboptimal T>MIC values and, possibly, spread of antimicrobial resistance. The lack of the perspective of age-dependent change in cefquinome clearance is a limitation of the study and should be described as such. Ref: Pozniak, B., Paslawska, U., Motykiewicz-Pers, K., & Switala, M. (2017). The influence of growth and E. coli endotoxaemia on amoxicillin pharmacokinetics in turkeys. British poultry science58(4), 462-468.)
  3. The manuscript has several language issues. I would recommend asking a native English speaker or a person very fluent in English and scientific writing to have a thorough look at the text. E.g. I would recommend using “administration” rather than “giving”. Please make sure to correct grammar issues, e.g. using adjectives and nouns, mixing countable and uncountable nouns, problematic suffixes.
  4. Title: I recommend to delete the word broiler and use only turkeys. In contrast to chickens, which are raised as either broilers or layers, turkeys are typically kept for meat (excluding the breeders). Therefore, by convention, it is recommended to refer to them simply as turkeys, not broiler turkeys.
  5. Raw data (individual concentrations, PK parameters and demographic details of the subjects) should be provided as a separate supplementary data. As far as I understand this is obligatory for the Journal.

Specific issues:

Abstract

  1. Line (L) 23: should read: “pharmacokinetic analysis”.
  2. L53: “primary renal elimination in an unchanged form”
  3. L59: “bacterial diseases” would sound more natural than “bacterial contagion” to my ear.
  4. L64: “have been studied extensively” or “extensive pharmacokinetic studies have been performed”
  5. L66: Keep the plural form
  6. L70: “limited amount of research has been”
  7. L72: What do you mean by “layer birds”? This is not a species. If you mean laying hens, please state so. The original cited paper keeps naming the study subjects as “layer birds” which is a very unfortunate expression as we know a lot about the study but still we don’t know what species it was performed in.
  8. L75-76: This statement only touches this important aspect which has a profound effect on the PK of drugs in turkeys. It obviously lacks references. The use of allometry in birds is proposed here: Lashev, L. D., & Haritova, A. M. (2012). Allometric analysis of antibacterial drugs in avian species. Bulgarian journal of veterinary medicine15(2). But another systematic assessment proved it to be prone do errors as discussed in here: Pozniak, B., Tikhomirov, M., Motykiewicz‐Pers, K., Bobrek, K., & Switala, M. (2020). The influence of age and body weight gain on enrofloxacin pharmacokinetics in turkeys—Allometric approach to dose optimization. Journal of veterinary pharmacology and therapeutics43(1), 67-78. The latter paper also indicates the intra-species scaling as a better use for allometry. Moreover, what is even more relevant for your study, it is also showing that the turkeys’ age will strongly affect the clearance of the drug and results obtained in only one age group cannot be considered as universal for turkeys of all ages. These studies do not deal with cefquinome but their conclusions were shown to be relevant for many other antimicrobials.
  9. L77: “in turkeys”
  10. L90: Please specify the line and give the body weight as mean, sd and range. Since the study was performed with the wash-out period in between, please specify these values for both time-points.
  11. L91: “turkey farm”; no “a” before humidity
  12. L96: signs of disease (no plural), “protocol was developed” sounds somehow better to me.
  13. L101: Dose can be calculated, not detected. This is not a cross-over study design. In a proper cross-over study you take a half of the flock and subject it to treatment A whereas the other half is simultaneously subjected to treatment B. Then, after some period (here, the wash-out period), you switch the groups and the half that got treatment A gets treatment B and vice versa. Again, this should happen simultaneously to avoid the confounding effect of time. Here is an exemplary reference: Sibbald, B., & Roberts, C. (1998). Understanding controlled trials crossover trials. Bmj316(7146), 1719-1720.

In your case this is particularly important because the body weight increase in turkeys is particularly rapid and the differences in the PK may not be attributed to the route of administration only, but also to the fact, that the cardiac output in turkeys that are three weeks older is simply different (lower as shown by Switala, M., Pozniak, B., Pasławska, U., Grabowski, T., Motykiewicz‐Pers, K., & Bobrek, K. (2016). Metronidazole pharmacokinetics during rapid growth in turkeys–relation to changes in haemodynamics and drug metabolism. Journal of veterinary pharmacology and therapeutics39(4), 373-380. Or Romvári, R., Petrási, Z., Sütő, Z., Szabó, A., Andrássy, G., Garamvölgyi, R., & Horn, P. (2004). Noninvasive characterization of the turkey heart performance and its relationship to skeletal muscle volume. Poultry science83(4), 696-700.).

This limitation has to be clearly stated in the manuscript.

  1. L105: Leg muscle? Please specify which. Additionally, please discuss the possible role of the renal portal system in the bioavailability of drugs administered in the caudofemoral portion of the body in birds. E.g. Frazier, D. L., Jones, M. P., & Orosz, S. E. (1995). Pharmacokinetic considerations of the renal system in birds: part I. Anatomic and physiologic principles of allometric scaling. Journal of avian medicine and surgery, 92-103. Obviously from your results the effect is rather moderate but you need to discuss it – the administration into the breast muscle might have led to even greater bioavailability.
  2. L113: Please rephrase the first sentence.
  3. L118-19: Please specify the technical details (including the manufacturer) of the HPLC system and the column.
  4. L125: Method validation should be a numbered subsection.
  5. L135: Please explain what was the method of LOD and LOQ determination. Rephrase the last sentence of this paragraph.
  6. L137: I see several issues here. You use a very old reference while more modern guidelines have been developed. Why haven’t you checked the non-specific binding? This can be performed easily by pre-treating the device with Tween. It is clear that it wasn’t high in your case as it would add up to the protein binding calculated in your study but this should be performed routinely to ascertain the validity of the results. Please comment on that as a limitation.
  7. L140: no capital letter needed, delete “way”
  8. L142: 30 min
  9. L153: I strongly suggest to switch to the more accepted presentation of data as mean and SD
  10. L154: These are parameters not variables.
  11. L154: What do you mean by the calculated mean CFQ concentrations? You need to calculate individual parameters from individual concentrations and not to pool them. Please rephrase for clarity.
  12. L159: AUC to infinity or last? What was the %AUCrest? BLOQ data were included or not? How was the elimination half-life calculated? (data points, acceptance criteria etc.)
  13. Table 1 simply repeats the data in Fig. 1 and is, therefore, redundant. Please provide the individual CT data as a separate file.
  14. L173: rephrase, e.g. “declined in a biphasic manner”
  15. L175: No need to repeat the full names of the parameters if you explained them in M&M section.
  16. L187: strange formatting
  17. Table 2. Specify the parameters as explained in point 31. Add AUCrest to allow the assessment of data quality.

Discussion

  1. L196: Rephrase the first sentence so that the acronym is first explained properly.
  2. L200: If you write about avian species please name the “layer birds” with their species name.
  3. L207: What do you mean by “beneficial disposition”? This has to be explained.
  4. L210 and throughout the manuscript: the Authors seem to use the word “animal” when they mean “mammal”. Please correct throughout the manuscript.
  5. L212: “at the indicates doses and routes.” Please rephrase for clarity.
  6. General point: please consider comparing the PK parameters obtained in other species in a tabular form. This will save time and space, and will facilitate comparisons for the reader.
  7. L222-223: You seem to repeat the “speedy elimination” too many times. Please be more concise.
  8. L224: “substantially longer”
  9. L228: delete redundant “also”
  10. L229-232: This sentence seem not to have much relevance to your work as you compare PK values of other drugs in fish. Not too much in common with turkeys and cefquinome. Moreover this sentence does not support the last sentence in this paragraph (L232-233).
  11. L234: Vdss is not a “rate”. Please rephrase.
  12. L235: no need for plural here (constants).
  13. L239: “tissue distribution” or “penetration to deeper compartments”. This is irrespective of the route of administration. You determine Vd after i.v. administration only for precision. The parameter describes the volume od distribution also if the drug is administered by other routes. However, in the latter case Vd is biased by F.
  14. L246-248: “These data of limited distribution indicate the speedy elimination of CFQ after IV giving in turkeys, and most other avian, animals, and fish species.” This is not true. Vd does not describe elimination, even though it may be affected by elimination. Most beta-lactams distribute in the extracellular space and this Vd is typical for such drugs. This is caused by the limited penetration of beta-lactams to the intracellular compartment. Moreover, you just mentioned that in fish the half-life can be up to 20 h. Please keep consistency in your reasoning.
  15. L249: lipophilicity would be a better term, or “solubility in lipids”
  16. L249: The opposite is the case. Low plasma protein binding typically allows better penetration to the extravascular space, particularly for drugs with relatively low Vd (ref. Toutain, P. L., & BOUSQUET‐MÉLOU, A. (2004). Volumes of distribution. Journal of veterinary pharmacology and therapeutics27(6), 441-453.)

Additionally, make sure you only compare Vdss and not Vz from other studies.

  1. L252: “value of CL”, not rate
  2. L264: How values of 0.020 and 0.037 L/h/kg may indicate faster clearance than values of 323 L/h/kg in your study or similar values in other studies? Please remember that i.p. administration, although allows for fast absorption, is not the same as i.v. administration.
  3. L266-267: This is nonsense. How interspecies differences can be explained by the same physicochemical nature of the drug you study? Additionally, you write about fish without giving protein binding in these species and later you write that the differences are due to limited binding capacity. Please keep consistency in your reasoning.
  4. L269: Please rephrase the text here. The absorption seems slower in turkeys than in the other species you mention as described by the longer absorption half-life.
  5. L274-275: “These data demonstrated that CFQ takes less time to enter the systemic circulation and establish an efficient plasma level in turkeys, other avian, animal, and fish species.” Less than what? Maybe you mean little time?
  6. L281: variable (no “s”). Why don’t you compare the elimination half-life after i.v. and i.m. administration?
  7. L285-295: These comparisons only make sense if the dose administered is the same. Please either provide this information or totally reconsider this paragraph.
  8. L297-311: Please consider the role of the renal portal system while assessing the bioavailability and comparing it with other studies. Similarly, formulation, site of injection all may play a role. Remember that F exceeding 100% needs attention and may be observed in formulations characterised with prolonged release. Beta-lactams do not easily permeate into cells which can be appreciated also from your estimate of Vd. Please reconsider the explanation of the high F in L308-9.
  9. L310: “famous” is not a good word choice here.
  10. L313: What do you mean by “often”? Please rephrase.
  11. L318: Please rephrase this unfortunate sentence. In the current form it suggests you do not understand the nature of MIC.
  12. L333-335: This recommendation may be valid only for turkeys of this given body weight. Quite likely, this dosage would lead to T>MIC less than 40% in young turkeys.
  13. Conclusions: Please rephrase the conclusions as the first sentence is hard to read.

Author Response

We appreciate the time and efforts that you spent on the extensive review of our manuscript. We believe that these comments and suggestions are from an expert in pharmacokinetics and substantially improved the quality of our manuscript. Thank you so much. We replied to your comments and suggestion here point by point and in the manuscript. Changes were done in red-colored font and the track function of the Word file is activated. We hope that our replies and editing of the manuscript meet your satisfaction.

Please find our response to your comments in the attached file and the Revised manuscript. 

Reviewer 2 Report

Manuscript entitled"Disposition of cefquinome in broiler turkeys (Meleagris gallopavo) following intravenous and intramuscular administrations" by Authors Mohamed Elbadawy et al describes about preclinical pharmacokinetic study of cefquinome in Turkeys. 

Comments: 

  1. Authors could have used an internal standard for the bio-analysis
  2. Line# 215: IV giving: please change as IV administration (many places)

  3. Discussion part: Line#286: Cmax of CFQ in turkey is less than Ducks (Reference#9). Please note that Ducks received 5mg/kg dose whereas, this study was done at 2mg/kg dose. When you compare, please mention the dose levels
  4. Plasma protein binding study concentration range 0.01 to 5ug/mL was provided. Please provide the actual concentrations studied and provide the results as table.
  5. Only 2 tables and one figure was provided. Authors could have provided more data from bio-analytical method validation (this could bring additional result tables and details)
  6. Authors have mentioned after 24h no CFQ was observed and suggested twice daily dosing, it would be appropriate if the authors could have studied higher dose or repeated dose after 12 hours
  7. When authors present bio-analysis validation, please consider providing representative chromatograms

Overall this manuscript needs some additional study. Whatever results are provided is good and additional data may be required.

Author Response

We appreciate the time and efforts that you spent on the extensive review of our manuscript. We believe that these comments and suggestions are from an expert in pharmacokinetics and substantially improved the quality of our manuscript. Thank you so much. We replied to your comments and suggestion here point by point and in the manuscript. Changes were done in red-colored font and the track function of the manuscript file is activated. We hope that our replies and editing of the manuscript meet your satisfaction.

Reviewer 3 Report

Abstract: Line 30, change giving to injection. Please do the same correction throughout the manuscript.

Method validation: Specify how the standard stocks were prepared. Since the stock and drug purity are different, please specify how the corrections were made for analysis.

Give the detailed specifications for the analytical column.

The internal standard for CFQ is widely available. Why was it not used in the bioanalysis of plasma samples?

2.5 How was the standard calibration curve prepared for the protein binding experiment?

Table 1 and 2: Please limit the values to 3 significant digits only.

Line 165: What is MAT?

Tmax is a categorical PK parameter. Please give the median and range for Tmax.

The discussion is very long. Please restructure and curtail some unnecessary statements.

Author Response

We appreciate the time and efforts that you spent on the extensive review of our manuscript. We believe that these comments and suggestions are from an expert in pharmacokinetics and substantially improved the quality of our manuscript. Thank you so much. We replied to your comments and suggestion here point by point and in the manuscript. Changes were done in red-colored font and the track function of the Manuscript file is activated. We hope that our replies and editing of the manuscript meet your satisfaction.

Reviewer 4 Report

The study of Elbadawy et al. aims to assess the pharmacokinetics of cefquinome (CFQ) in broiler turkeys after single IV and IM administration of the drug at a dose of 2 mg/kg of body weight. The Authors employed non-compartmental analysis to calculate pharmacokinetic parameters of the studied compound. Subsequently, they discuss and compare obtained results with the literature data that include pharmacokinetic parameters of CFQ in various species. The results indicate that the administered dose of 2 mg/kg is sufficient to achieve the therapeutic effect (MIC for several bacterial strains). This study may be useful for readers, since PK of CFQ has not yet been studied in broiler turkeys. However, it requires some corrections and modifications before it will be suitable for publication in Pharmaceutics.

Major points;

  1. In non-compartmental analysis, it should be explained how AUC or AUMC was calculated (from time 0 to infinity or to the last measured point)
  2. If Authors have concentration-time data for each animal, they should calculate PK parameters for each individual animal and then calculate the mean and the standard deviation. This would give some information on the between-subject variability in PK parameters.
  3. The PK analysis performed in this study is rather limited. Only non-compartmental approach was used to calculate PK parameters. It would be very useful to perform a compartmental analysis and estimate PK parameters using non-linear regression. This approach enables to perform simulations of multiple dosing. In line 334 the Authors claim that twice-daily dosage of 2 mg/kg is suitable. It should be confirmed by 1) estimating PK parameters using compartmental analysis, 2) preforming simulations of multiple dosing, 3) comparing results of simulation with MIC for several susceptible bacterial strains. This may be performed in WinNONLIN program, which was used in this investigation.
  4. In the discussion section, the Authors compare PK parameters of CFQ in turkeys with values obtained in previous publications using various species. It would be much clearer and more convenient for readers if Authors provided a table in which they collect all the parameters for different species.
  5. The Authors should not repeat so frequently in the text of the manuscript the values of PK parameters that are shown in tables. Also, please make sure that the values of parameters are correct. In line 176, the value of Vdss is slightly different than that presented in table 2.

Minor points:

  • line 120 – please provide the concentration of the buffer and flow rate. Please provide details regarding the HPLC system and the column (manufacturer, model and software used in analysis).
  • line 158 – ‘trapezoids method’ should be ‘trapezoidal method’
  • line 234 – it cannot be said that ‘Vdss rate represents the diffusion extent of a drug in the body tissue’. 1) This is not ‘rate’ 2) passive diffusion is not the only mechanism engaged in drug distribution.
  • 256 – clearance cannot be ‘quick’. We can say that it is large or small.
  • line 154, 156 – not ‘pharmacokinetic variables’ but ‘pharmacokinetic parameters’
  • line 136 -’ The CFQ peak in plasma was appeared at 12 minutes’ the phrase in misleading. It is not clear whether the Authors mean Cmax value in plasma or a peak on a chromatogram.
  • The manuscript should be corrected by a native English speaker or by a professional language editing service.

Author Response

(The authors gave the same response as above.)

Round 2

Reviewer 1 Report

Dear Authors,

Thank you for the corrections you have made. Thanks to your efforts the manuscript looks significantly better now. There are, however, still some things I would like to ask you to fix:

  1. Abstract: "plasma protein binding of CFQ" (no need for "percentage")
  2. L78: typo in CFQ
  3. L131: Missing value
  4. L156: Current form does not specify the concentration at which the drug protein binding was performed at. Only the concentrations in deionised water are mentioned but no information on the final plasma concentration. Additionally, was it fresh or frozen plasma?
  5. You still did not provide the method of determination of the terminal half-life in the PK section of the M&M. Please amend it. Additionally, write how many time-points have been used. If many, specify what was the minimum number. Remember that Cmax cannot be included. Elimination rate constant (kel) is not the same as beta. If you calculate beta, it suggests you used the compartmental approach and observe a biexponential decay. Under such conditions, beta is not k10 which equals (alfa x beta)/k21. Please refer to the standard PK handbooks (e.g. Riviere). Terminology in different PK frameworks needs to be clearly identified. If you haven't fitted a biexponential model and just assessed the slope of the elimination phase you can write that you determined the kel but in such case you need to specify how many points you have used or what were the criteria of the time-point choice for fitting the regression line.
  6. L189 "are presented in Supplementary Table 2" (no parenteses).
  7. L190: concentrations declined (no "were")
  8. L212 no "was"
  9. Table 2. Correct the unit (μg/mL)
  10. L218L This repetition doesn't make much sense. I suggest deleting the whole part starting with "which" and ending with "elimination".
  11. L221 delete "as"
  12. L229: You still did not explain what you mean by favourable this time. Do not look for synonyms but rather tell directly what you mean. Rapid and almost complete absorption? Satisfactory elimination rate that allows reasonable dosing intervals? Be more specific.
  13. L234: delete "after administration" 
  14. You are still using animals instead of mammals in several points. Please amend it.
  15. L278: Be more specific: "The protein binding assay used in this study did not account for the non-specific drug binding to the centrifugal device which may be a limitation in the present study"
  16. Table 3. I was rather expecting this table to replace the tedious comparisons in the text and not to supplement it. However, it's still better than the earlier version (just the text).
  17. L347: "It takes little time for CFQ to enter the systemic circulation"
  18. L351-352: This is incomprehensible, please correct it.
  19. L357: at a dose
  20. L368: SD of 0.00 looks strange, perhaps better to remove it.
  21. L386: This F value indicates almost complete absorption. You shouldn't expect much more. In my opinion the sentence should rather go: "High bioavailability of CFQ after IM administration into thigh muscles may suggest that the renal portal system and tubular excretion play an insignificant role in the potential first-pass effect often associated with the administration to the caudofemoral portion of the body in birds and reptiles (reference). However, lack of comparative data on CFQ administration into another part of the body (e.g. breast muscle) prevents any firm conclusions. This may be considered another limitation of the study."
  22. L397-398: The sentence still sounds poor. Perhaps something like this: "However, MIC values of ≤0.1 μg/mL typically reported for pathogens isolated from different species suggest high clinical efficacy of this antibiotic."
  23. L404: "fowls" is not a species
  24. L429" "...absorption and bioavailability."
  25. L430: C max is a point, not a a period that can be prolonged. Please rephrase. (e.g. "concentrations exceeding MIC values"
  26. L434: Rephrase, e.g.  "...and its residues in the edible tissues of this species"

Author Response

Dear Reviewer, we would like to thank you so much for improving our manuscript.

Please find the reply to comments in the attached file and at the corresponding areas of the manuscript (Trach changes is activated)

Reviewer 2 Report

Authors have addressed most of the comments and the manuscript looks improved now.

Authors need to consider the additional studies in future experiments. 

Author Response

Comments and Suggestions for Authors

-The authors have addressed most of the comments and the manuscript looks improved now.

-We would like to thank you so much for reviewing and improving of our manuscript.

-Authors need to consider the additional studies in future experiments. 

-Response: thanks for the valuable advice. We will consider the additional studies in our future experiments.

Reviewer 4 Report

The manuscript has been substantially improved according to the reviewers' suggestions and now it is suitable for publication.

- Please add the value of flow rate in the Methods section (line 131).

Author Response

Comments and Suggestions for Authors

-The manuscript has been substantially improved according to the reviewers' suggestions and now it is suitable for publication.

-The authors would like to thank you so much for reviewing and improving our manuscript.

- Please add the value of flow rate in the Methods section (line 131).

-Response: Thanks for your valuable notice. We added the value.

This manuscript is a resubmission of an earlier submission. The following is a list of the peer review reports and author responses from that submission.